# Price Competition and Product Differentiation Based on the Subjective and Social Effect of Consumers’ Environmental Awareness

**DOI:** 10.3390/ijerph17030716

**Published:** 2020-01-22

**Authors:** Dayi He, Ximing Deng

**Affiliations:** School of Economics & Management, China University of Geosciences, Beijing 100083, China; 2007180065@cugb.edu.cn

**Keywords:** price competition, product differentiation, consumers’ environmental awareness

## Abstract

Consumer environmental awareness (CEA) can affect green consumption decisions in different and confusing ways. In order to explain the reasons for these divergences, this study divides CEA into two main components: the subjective effect and the social effect. Then, we integrate the two effects into the classic Hotelling model to study the influence of CEA’s subjective effect and social effect on price competition and product differentiation strategy. It was found that the subjective and social effects of CEA have opposite impacts on price competition and product differentiation strategies. The subjective effect of CEA increases the price and profit level of enterprises, and enlarges the difference in the environmental friendliness of products. Meanwhile, the social effect of CEA reduces the enterprises’ price and profit level, and narrows the difference in the environmental quality of products. Therefore, we suggest that it is necessary for producers of green products to distinguish between these two effects. Numerical examples are provided to verify our findings. Finally, some possible suggestions regarding the competition of green products are put forward which take into consideration the subjective and social effects of CEA. The main contribution of this paper is to theoretically explain the opposite effects of the two different components of CEA on environmentally friendly product pricing and differentiation strategy; this presents a possible explanation as to why the behavior regarding CEA differs, and provides theoretical support for enterprises to price and differentiate green products.

## 1. Introduction

Consumer environmental awareness (CEA) has become an important issue affecting the consumers’ daily consumption decisions. A growing number of customers have adjusted their consumption preferences because of the influence of CEA [1,2], by choosing environmentally friendly products and preferring eco-conscious organizations [3,4,5]. In 2014, a Eurobarometer investigation on the environment in the 28 member states of the European Union showed that 75% of Europeans prefer to pay more for environmentally friendly products, up from 72% in 2011 [2]. In Ref. [6], it was found that CEA affects consumers’ purchasing decisions and behavior. In Ref. [7], it was revealed that CEA affects their willingness to buy green products. In Ref. [8], it was concluded that a large number of customers are willing to buy environmentally friendly products and pay more for products/services to show their increased environmental awareness and preference for green products. Empirical studies have also confirmed that consumers are willing to pay a premium for green products, owing to the additional utility they achieve from purchasing such products [7,9,10,11,12].

This shift in consumer consumption tendency provides new possibilities for competition among enterprises. In [13], it was argued that green marketing has created new opportunities for market development, differentiation, cost advantage, niche building, and customer segmentation. Understanding the consumers’ motivation to buy green products and the influencing factors allows one to formulate suitable strategies for developing markets [14], and helps in eliminating the obstacles to green consumption [15]. Hence, with the increasing willingness of consumers to buy green products, enterprises have to consider suitable price competition and product differentiation strategies to ensure their survival and development. Therefore, this study investigates the effects of CEA on environmentally friendly product pricing and differentiation decisions.

In pricing strategies for environmentally friendly products, CEA is commonly considered in the analysis of product choice and demand. However, the effect of CEA on consumer behavior is debatable [16,17]. In Ref. [18], it was shown that the higher the CEA level of consumers, the more willing they are to pay higher prices for environmentally friendly products, which provides support for the pricing strategy of green products. However, in Ref. [19], it is argued that CEA’s impact on consumer behavior was not entirely positive. They found that consumers were unwilling to trade convenience, quality, and other commodity characteristics for the benefits of green products. In Ref. [20], a large-scale review was conducted to find that there are more than 20 consumer-level theories grouped into six categories.

All these arguments are reasonable. As a matter of fact, the composition of CEA is complex. This might be the reason for the complexity of the impact of CEA on consumer behavior. When some aspects of CEA are of significance, their impact on consumer behavior may be positive; while when other aspects are significant, their impact on consumer behavior may be reversed. Therefore, if we take CEA as a whole when analyzing its impact on consumer behavior, we will have such divergence. In order to clarify the reasons for the differences in the impact of CEA on consumer behavior, this paper does not regard CEA as a whole, but divides it into two parts: the subjective and social effect. Then, we analyze the influence of CEA on pricing strategy and product environmental quality differentiation strategy.

## 2. Literature Review

Product differentiation is a marketing strategy that businesses use to distinguish a product from similar offerings on the market. The difference could be something concrete, like speed, power, performance, and better service, or, it could be a more ephemeral quality, such as just being cooler or more stylish than the competitors. There are three types of product differentiation—vertical, horizontal, and mixed differentiation. Vertical differentiation exists when consumers compare a product according to one feature—quality. Horizontal differentiation refers to distinctions in products that cannot be easily evaluated in terms of quality. This stands in contrast to vertical differentiation, where the distinctions between products are objectively measurable and are based on the products’ respective level of quality. Mixed differentiation is a combination of both vertical and horizontal differentiation. It is common when consumers consider more complex products or marketing executives are looking at more sophisticated markets.

With the development of environmental protection awareness, an increasing amount of research is focusing on product pricing and differentiation based on CEA differences. This paper mainly focuses on the related theoretical research. In the model developed in [21], firms based their differentiation strategies on the environmental quality of products. They found that consumers with lower environmental quality requirements had lower willingness to buy differentiated products. In Ref. [22], a vertical differentiation model was built to investigate the provision of environmental quality in an imperfectly competitive market. In a duopoly model with vertical product differentiation, Ref. [23] investigated firms’ product strategy in the case of various environmental policy instruments. In Ref. [24], a vertically differentiated model was established in an imperfectly competitive market, where consumers are environmentally aware. In Ref. [25], they propose a model of horizontal product differentiation combined with the consumption externality to study the market implication of CEA’s externalities. In Ref. [26], a model with one domestic and one foreign firm located at each end of the 0–1 product line is presented, where products are differentiated according to both environmental quality (vertical differentiation) and taste/eco-label (horizontal differentiation). In [27], they developed a horizontal product differentiation model, where two firms are also located at different ends of a quality scale. In Ref. [28], they established a spatial duopoly model to study the effects of environmental concern on competing firms’ decisions regarding prices, product characteristics, and market shares. They found that the market share of the environmental goods increases in each type with environmental concern, and it declines if costs are higher to produce these goods. Their research framework regarding CEA’s effect influenced other studies, such as [29,30,31,32]. In Ref. [33], CEA is depicted from two aspects: the scale of green consumers and the extra cost they are willing to pay for green products. They discussed the role of CEA in product competition between traditional producers and green producers.

All the above studies take CEA as a whole and integrate it into the settings of the model. Although different ways of differentiation are used, the conclusions are different. This study also considers the effects of CEA in green product pricing and differentiation. However, it varies from the earlier research in that the effects of CEA are divided into two parts: the subjective and social effects, instead of dealing with the impact of CEA as a whole.

Empirical studies have confirmed that subjective norms and social impacts are considered as important factors to explain the consumers’ attitude–behavior relation in the field of green consumption. Subjective norms are a reflection of the social pressure that is perceived by the person and shapes a certain behavior [34]. In other words, subjective norms indicate the person’s perception of whether people decide a certain behavior or not, i.e., it is the personal perception of social-norm pressure or the beliefs of others that determine whether the person should behave in a certain way or not. Thus, subjective norms depend on a personal knowledge of prominent opinions being accepted by the person [1]. Studies in the field of psychology suggest the theory that subjective norms are an important factor in behavioral intention. Social impacts are the degree of influence of family and relatives, such as parents, relatives, acquaintances, and close friends on the behavior and decisions of each individual. According to the prevailing theory, social norms often bring pressure to individuals. When individuals deviate from social norms, this pressure will promote individuals to return to the requirements of social norms to a certain extent. Moreover, the higher the degree of deviation, the stronger the pressure. In Ref. [35], it is shown that subjective norms, such as attitudes towards buying organic food and perceived behavioral control, have a strong impact on intentions among consumers. In Ref. [36], it was found that personal norms are the second important predictor of green purchasing behavior. In Ref. [37], it is shown that consumers tend to act consistently with the social norms, either to maintain their social attachment or to keep their self-congruity.

The effect of CEA on consumer behavior provides firms with the opportunities to differentiate products and to diversify their competition strategies. In the traditional the competitive environment, low prices are often attractive to consumers, because the characteristics of these products are similar. Moreover, in this case, price competition has become the main means for enterprises to gain a market share, and the competition among enterprises is quite fierce. However, in the field of green consumption, consumers will accept green products with higher prices because they recognize that green products have higher production costs [28]. This recognition of the cost of green products can provide a new opportunity for enterprises besides pure price competition.

This paper aims to analyze product differentiation strategies based on CEA. On the basis of the standard Hotelling linear city model, we develop a model of price competition and product differentiation incorporating the subjective and social effects of CEA. The model consists of two firms and a continuum of potential consumers represented by the unit interval. Our analysis is based on a two-stage noncooperative game. In the first stage, the two firms determine the level of environmental friendliness of their products; in the second stage, the two firms decide the price of their products at the same time.

We modified two factors in the standard Hotelling model, where the distance is used to differentiate products, and the cost of transportation represents the cost/utility that a consumer would pay for the difference of the product. However, one may buy green products at the cost of two factors: subjective and social norms. Hence, we change the consumers’ utility function to incorporate the two costs. Besides, the marginal cost in the Hotelling model is the same, even being normalized to zero sometimes. However, firms produce greener products at a higher cost. In this case, the marginal cost cannot be neglected. Therefore, the marginal cost in our model is set to be a function of the environmental friendliness.

The rest of this paper is organized as follows. A game-theoretical model is established in Section 3. The solutions are derived and analyzed in Section 4. In Section 5, numerical examples are presented to verify our results. Then, some recommendations for price competition and product differentiation strategies are proposed in Section 6. The final section (Section 7) concludes this study.

## 3. The Model

Assume two firms compete in terms of product prices and environmental friendliness. There is a continuum of consumers uniformly distributed over the interval [0,1]. The game includes two stages. In the first stage, the firms simultaneously choose the environmental quality of products. In the second stage, the two firms compete in terms of price after observing the rival’s choice. At this stage the product characteristics are determined and irreversible, so that price competition is influenced by the degree of product differentiation. The two firms sell a heterogeneous product with characteristics *a* and 1−b, where 0≤a, b≤1. For simplicity, assume that a+b≤1. When a=b=0, the two firms locate at the left and right end and they provide products with the largest difference in terms of environmental friendliness. When a+b=1, the two firms are located at the same point which means they provide products with the same level of environmental friendliness.

The consumer’s willingness to pay is determined by their intrinsic utility, the level of this CEA, product characteristics, and social expectation regarding CEA. Let’s assume that one consumer’s willingness to pay with the level of CEA at θ∈[0,1] for one unit of the product of quality *x* is defined by
(1)$$v\left(x,\theta \right)=v_{0}-t{\left(x-\theta \right)}^{2}-d{\left(x-x_{s}\right)}^{2},$$
where $v_{0}$ stands for the gross intrinsic utility a consumer derives from consuming one unit of the product. Similar to the classic Hotelling model, assume that $v_{0}$ is sufficiently large to ensure that every consumer buys one product. The term $t{\left(x-\theta \right)}^{2}$ represents the cost a consumer, located at θ, pays if they do not get their preferred product environmental quality because they have to buy from firm *i* selling the product at characteristic *x*. Hence, this term can be regarded as the loss of utility a consumer would pay due to the subjective effect of CEA. The term $d{\left(x-x_{s}\right)}^{2}$ represents the social effect of CEA on one consumer when they deviate from the social expected level of environmental friendliness $x_{s}$.

Assume that the consumer located at $\hat{\theta }$ is indifferent in terms of buying from the two firms. Therefore, consumers located on the left side of $\hat{\theta }$ will buy from Firm 1 and the rest will buy from Firm 2. In addition, we define the socially expected level of environmental friendliness $x_{s}$ as the average level of environmental friendliness of all the sold products, i.e.,
(2)$$\bar{x}=\int_{0}^{\theta }af\left(x\right)dx+\int_{\theta }^{1}\left(1-b\right)f\left(x\right)\;dx=a\theta +\left(1-b\right)\left(1-\theta \right).$$

This assumption is different from that in [28]. They fix the social expectation level to 1, which is the highest value of product environmental quality. As a matter of fact, the social expectation usually is not fixed at the highest level. Instead, it is more of an average and changeable. Moreover, the environmental quality level of social expectations is affected not only by the environmental quality level of the products provided by enterprises, but also by factors such as enterprise publicity and consumer perception. Therefore, it is dynamic and should not be fixed. In this case, we set the social expectation level of environmental quality of products as the average level of environmental quality of products purchased by all consumers, which integrates the influence of producers and consumers. On the basis of this assumption, one part of consumer expectation in terms of the environmental quality of products is higher than the social expectation; the other part is lower. Such that, this assumption provides an opportunity for enterprises to apply price and product differentiation strategies based on the two components of CEA.

If one consumer located at $\hat{\theta }$ buys from Firm 1, then their utility is
(3)$$U_{1}\left(a,\theta \right)=v_{0}-t{\left(a-\theta \right)}^{2}-d{\left(a-\bar{x}\right)}^{2}-p_{1}.$$

Otherwise, they buy from Firm 2, and their utility is
(4)$$U_{2}\left(b,\theta \right)=v_{0}-t{\left(\left(1-b\right)-\theta \right)}^{2}-d{\left(\left(1-b\right)-\bar{x}\right)}^{2}-p_{2},$$
where $p_{i}$ is the price for the product set by the firm *i*, respectively.

Therefore, $\hat{\theta }$ must satisfy the following equation:
$$U_{1}\left(a,\hat{\theta }\right)=U_{2}\left(b,\hat{\theta }\right).$$

Such that
(5)$$\begin{array}{cc} \hat{\theta }= & \frac{p_{2}-p_{1}}{2(1-(a+b))(t-d(1-(a+b)))}\\ \; & +\frac{t(a+(1-b))}{2(t-d(1-(a+b)))}-\frac{d(1-(a+b))}{2(t-d(1-(a+b)))}. \end{array}$$

Then, consumers with $\theta <\hat{\theta }$ will buy the product of Firm 1 and consumers with $\theta >\hat{\theta }$ will buy the product from Firm 2. Therefore, on the basis of the uniform distribution of consumers, the demand functions for Firm 1 and Firm 2 are
(6)$$\begin{array}{cc} S_{1}=\hat{\theta }= & \frac{p_{2}-p_{1}}{2(1-(a+b))(t-d(1-(a+b)))}-\frac{d(b-a)(1-(a+b))}{2(t-d(1-(a+b)))}+a+\frac{1-b-a}{2}\\ = & \frac{p_{2}-p_{1}}{2D(t-dD)}-\frac{d(b-a)D}{2(t-dD)}+a+\frac{D}{2} \end{array}$$
(7)$$\begin{array}{cc} S_{2}=1-\hat{\theta }= & \frac{p_{1}-p_{2}}{2(1-(a+b))(t-d(1-(a+b)))}+\frac{d(b-a)(1-(a+b))}{2(t-d(1-(a+b)))}+b+\frac{1-b-a}{2}\\ = & \frac{p_{1}-p_{2}}{2D(t-dD)}+\frac{d(b-a)D}{2(t-dD)}+b+\frac{D}{2}, \end{array}$$
where D=(1−b)−a is the difference between the two firms in terms of environmental friendliness of products. In the classic Hotelling model, the equilibrium result is symmetrical, where the two firms locate at each end of the city, and they split up the market at the same price, that is a=0,b=0, $p_{1}=p_{2}$, and $S_{1}=S_{2}=1/2$. This result also exists in (Equation 6) and (Equation 7) obviously. In the sales function (Equation (Equation 6)) of Firm 1, the first item is the effect of the subjective and social norms of CEA, and the difference of price ($p_{2}-p_{1}$) on sales. It indicates that the price difference will affect the market share of the two firms. However, this impact is adjusted by the environmental friendliness of the enterprise’s products and the subjective and social impact of CEA. The second item represents the influence of the subjective and social effect on the market share when the two firms choose different levels of environmental friendliness. It mainly depends on the difference of b−a adjusted by the subjective and social impact of CEA. Firm 1 (Firm 2) will lose a part of market to Firm 2 (Firm 1) if a≤b (a≥b). The third term is the territory of Firm 1. The last item is the average value between Firm 1 and 2. The last two items are consistent with the classic Hotelling model. Consumers at left side of *a* will definitely buy from Firm 1, and consumers between *a* and 1−b are divided equally by the two firms. Obviously, the sales function of Firm 2 has a similar structure.

Environmentally friendly products incur production higher costs. Hence, we assume that the cost to produce one product is positively proportional to the level of the environmental friendliness. Profits of the two firms are then defined as
(8)$$\begin{array}{cc} \pi_{1} & =\left(p_{1}-ca\right)S_{1} \end{array}$$
(9)$$\begin{array}{cc} \pi_{2} & =\left(p_{2}-c\left(1-b\right)\right)S_{2}, \end{array}$$
where c>0 is the marginal cost in terms of the level of environmental friendliness. From the FOCs for a profit-maximizing price strategy, we obtain the following equilibrium prices:
(10)$$\begin{array}{cc} p_{1} & =ac+\left(1-\left(a+b\right)\right)\left[\frac{t}{3}\left(3+a-b\right)-d\left(1-\left(a+b\right)\right)+\frac{c}{3}\right] \end{array}$$
(11)$$\begin{array}{cc} p_{2} & =\left(1-b\right)c+\left(1-\left(a+b\right)\right)\left[\frac{t}{3}\left(3-a+b\right)-d\left(1-\left(a+b\right)\right)-\frac{c}{3}\right]. \end{array}$$

It can be found that prices consist of two parts. The first item is the cost to produce a single unit at the level of *a* or 1−b of the environmental friendliness. The second item is the difference in the two firms’ product in terms of the environmental friendliness, which is adjusted by the subjective and social effect of CEA. Then, by substituting (Equation 10) and (Equation 11) into (Equation 6) and (Equation 7), we can obtain the firms’ sales volume at equilibrium prices:
(12)$$\begin{array}{cc} S_{1} & =\frac{t(3+a-b)-3d(1-(a+b))+c}{6(t-d(1-(a+b)))}=\frac{1}{2}-\frac{(b-a)t-c}{6(t-d(1-(a+b)))} \end{array}$$
(13)$$\begin{array}{cc} S_{2} & =\frac{t(3-a+b)-3d(1-(a+b))-c}{6(t-d(1-(a+b)))}=\frac{1}{2}+\frac{(b-a)t-c}{6(t-d(1-(a+b))).} \end{array}$$

If the two firms are in a symmetrical position, they will share the market equally. However, when the subjective and social norms of CEA are taken into consideration, the market share will deviate from the middle point at the degree presented by the second part in Equations (Equation 12) and (Equation 13). Then, by substituting (Equation 10) and (Equation 11) into (Equation 8) and (Equation 9), we can obtain the firms’ profits at equilibrium prices:
(14)$$\begin{array}{cc} \pi_{1} & =\frac{\left(1-\left(a+b\right)\right){\left(t\left(3+a-b\right)-3d\left(1-\left(a+b\right)\right)+c\right)}^{2}}{18(t-d(1-(a+b)))} \end{array}$$
(15)$$\begin{array}{cc} \pi_{2} & =\frac{\left(1-\left(a+b\right)\right){\left(t\left(3-a+b\right)-3d\left(1-\left(a+b\right)\right)-c\right)}^{2}}{18(t-d(1-(a+b)))}. \end{array}$$

Obviously, the two firms’ profits are non-negative.

Then, we come to the first stage of the game. In this stage, the firms choose their level of environmental friendliness to maximize profits. On the basis of (Equation 14) and (Equation 15), we can solve the following equations:
(16)$$\frac{\partial \pi_{1}}{\partial a}=0,\;\frac{\partial \pi_{2}}{\partial b}=0.$$

From these equations, we can obtain three solutions, which are discussed in the next section.

## 4. Results

There are three solutions to (Equation 16). Two of them are degenerate equilibrium states because only one firm survives in the market, which results in no price competition and product differentiation. Hence, we do not focus on the two solutions (in Section 4.1). Instead, we concentrate on the last normal solution.

### 4.1. Two Degenerate Equilibrium States

The first solution of (Equation 16) is
(17)$$a=\frac{1}{2}-\frac{c}{2t}-\frac{3t}{3d+t},\;b=\frac{1}{2}+\frac{c}{2t}$$

As for 0≤b≤1, such that $0\leq \frac{c}{t}\leq 1$. Additionally, 0≤a+b≤1, then
$$0\leq \frac{3t}{3d+t}\leq 1\Leftrightarrow \frac{d}{t}\geq \frac{2}{3}.$$

Hence, Equation (Equation 17) could be a Nash equilibrium if both $0\leq \frac{c}{t}\leq 1$ and $\frac{d}{t}\geq \frac{2}{3}$ hold.

Then, by substituting (Equation 17) into (Equation 10) to (Equation 15), the prices, sales, and profits of the two firms can be obtained.
(18)$$\begin{array}{cc} p_{1} & =\frac{c}{2}\left(1-\frac{c}{t}\right)-\frac{3ct}{3d+t}=ac=0 \end{array}$$
(19)$$\begin{array}{cc} p_{2} & =\frac{c}{2}\left(1-\frac{c}{t}\right)+\frac{6t^{3}}{{\left(3d+t\right)}^{2}}=\left(1-b\right)c+\frac{6t^{3}}{{\left(3d+t\right)}^{2}} \end{array}$$
(20)$$S_{1}=0,\;S_{2}=1$$
(21)$$\pi_{1}=0,\;\pi_{2}=\frac{6t^{3}}{{\left(3d+t\right)}^{2}}.$$

Furthermore, we can derive that
$$\theta =0,\;\bar{x}=\frac{1}{2}-\frac{c}{2t}.$$

Hence, the Nash equilibrium is
(22)$$a=0,\;b=\frac{1}{2}+\frac{c}{2t}.$$

In this case, Firm 1 withdraws from the market competition (providing ordinary products), and Firm 2 monopolizes the whole market. It is noted that
(1)Although Firm 2 occupies a monopoly position, it does not set the level of environmental friendliness of products to 0. In that case, both Firm 1 and 2 produce ordinary products, thus entering the traditional price competition situation. It can be found that the difference between Firm 1 and 2 in the degree of environmental friendliness of products is $\left(1-b\right)-a=\frac{1}{2}-\frac{c}{2t}$, which is negatively related to marginal production costs and positively related to the subjective effect of CEA, but not related to the social effect of CEA.(2)The price of Firm 1 is 0. The price of Firm 2 is composed of the unit production cost and the premium. The higher the unit production cost, the higher the firm pricing. The premium is determined by the subjective and social effects of CEA. It is an increasing function of the subjective effects and a decreasing function of the social effects.(3)In terms of profits, because Firm 1 has no sales volume, its profit is zero. Firm 2 monopolizes all market demand and its sales volume is 1, so its profit is the premium part.


From the above results, we can propose that
(1)When the rival is incapable or unwilling to provide green products, the optimal decision of the firm is to set its environmental friendliness of products at the level of $\frac{1}{2}-\frac{c}{2t}$, which is sufficient to completely monopolize the market. However, since pricing is an incremental function of marginal production costs, in order to avoid giving potential competitors the opportunity to enter as a result of overpricing, the firm should increase the research and development of environmentally friendly production, and effectively control the production cost of green products to maintain the advantages of price competition.(2)Furthermore, because premium (profit) is an increasing function of the subjective effect and a decreasing function of the social effect, the firm should emphasize the role of the subjective effect of CEA and the importance of green product characteristics in meeting the subjective effect of CEA (to increase *t*).(3)On the other hand, aiming to reduce the role of the social effects of CEA, through advertising, publicity, and other means, in order to enable consumers to rationally choose green consumption, without blindly pursuing the level of social expectations (to reduce *d*). However, in this case, the product characteristics provided by Firm 2 are the level of social expectations, so Firm 2 does not need to excessively emphasize to consumers how environmentally friendly its products are, so as to avoid the consumers’ excessive loss of consumption experience caused by inconsistencies with the level of social expectations.


The second solution is
(23)$$a=\frac{1}{2}-\frac{c}{2t},\;b=\frac{1}{2}+\frac{c}{2t}-\frac{3t}{3d+t}.$$

It can be shown that this solution could also be a Nash equilibrium if both $0\leq \frac{c}{t}\leq 1$ and $\frac{d}{t}\geq \frac{2}{3}$ hold. Similarly, by substituting (Equation 23) into (Equation 10) to (Equation 15), the prices, sales, and profits of the two firms can be obtained. It turns out that this equilibrium is just the symmetry of the first solution. In this case, Firm 1 monopolizes the market, while Firm 2 withdraws from competition.

In fact, the two solutions above are special cases where one firm survives in the market, which deviates from the focus of this paper. In the next section, we mainly analyze the third solution.

### 4.2. The Normal Result

The third solution is
(24)$$a=\frac{1}{2}-\frac{c}{2t}-\frac{3t}{4(3d+t)},\;b=\frac{1}{2}+\frac{c}{2t}-\frac{3t}{4(3d+t)}.$$

First, we show that (Equation 24) can be a feasible Nash equilibrium.

Let $D=\frac{3t}{2(3d+t)}$. If (Equation 24) is a Nash equilibrium, it must satisfy the following condition: a≥0, b≥0, and a+b≤1. It can verified that the last condition holds naturally because *t* and *d* are positive. From a≥0, we can obtain that D≤1−c/t. Meanwhile, it can also be derived that D≤1+c/t from b≥0. Such that we can obtain that D≤1−c/t, that is
$$\frac{3t}{2(3d+t)}\leq 1-\frac{c}{t},$$
such that
$$t^{2}-2\left(3d-c\right)t+6cd\leq 0.$$

This is an open-up quadratic function with regard to *t*, which can be proved to have two positive real roots when
$$\begin{array}{cc} \left(3d-c\right)-\sqrt{{\left(3d-c\right)}^{2}-6cd}\leq & t\leq \left(3d-c\right)+\sqrt{{\left(3d-c\right)}^{2}-6cd},\\ d & \geq \frac{2+\sqrt{3}}{3}c. \end{array}$$

To sum up, Equation (Equation 24) could be a Nash equilibrium when the above conditions are satisfied.

In this case, by substituting (Equation 24) into (Equation 10) and (Equation 11), we can obtain the equilibrium prices of the two firms as
(25)$$\begin{array}{cc} p_{1} & =\frac{c}{2}\left(1-\frac{c}{t}\right)+\frac{3t(2t^{2}+\left(3d-c\right)t-3cd)}{4{\left(3d+t\right)}^{2}}=ac+\frac{3t^{2}\left(3d+2t\right)}{4{\left(3d+t\right)}^{2}} \end{array}$$
(26)$$\begin{array}{cc} p_{2} & =\frac{c}{2}\left(1-\frac{c}{t}\right)+\frac{3t(2t^{2}+\left(3d+c\right)t+3cd)}{4{\left(3d+t\right)}^{2}}=\left(1-b\right)c+\frac{3t^{2}\left(3d+2t\right)}{4{\left(3d+t\right)}^{2}}. \end{array}$$

Furthermore, by substituting (Equation 24) into (Equation 12) and (Equation 13), we can obtain the equilibrium sales of the two firms as
(27)$$S_{1}=\frac{1}{2},\;S_{2}=\frac{1}{2}.$$

By substituting (Equation 24) into (Equation 14) and (Equation 15), we can obtain the equilibrium profits of the two firms as
(28)$$\pi_{1}=\pi_{2}=\frac{3t^{2}\left(3d+2t\right)}{8{\left(3d+t\right)}^{2}}.$$

From the above results, apart from the non-negativity of prices, sales, and profits, we can find the following properties as well.

(1) The pricing strategies of the two firms are the same because their prices have a similar structure. In the price composition, the first item is the production cost of a single unit under the level of environmental friendliness determined by the firm, which is proportional to the level of environmental friendliness. This means that when a firm chooses a higher level of environmental friendliness, it must adopt a higher price to ensure its profits. The second item, denoted as *P*, is identical for both firms. It can be regarded as a premium unit and a source of profits for the firms. First of all, the premium unit is completely determined by the subjective and social effects of CEA. Secondly, since
$$\frac{\partial P}{\partial t}\geq 0,\;\frac{\partial P}{\partial d}\leq 0.$$

Thus, the premium is positively related to the subjective effect of CEA and negatively related to the social effect of CEA. Therefore, in order to obtain more profits, enterprises should make full use of the subjective effects of CEA, and effectively restrain the social effects of CEA.

(2) The two firms share the market equally. Because $S_{1}=S_{2}$, it is easy to derive that the marginal consumer locates at the midpoint, i.e., $\hat{\theta }=1/2$. This means that half of consumers buy from Firm 1 and half of consumers buy from Firm 2. Hence, the average value of products’ environmental friendliness is
$$\bar{x}=\frac{1}{2}\left(1-\frac{c}{t}\right).$$

It can be found that the average decreases as the marginal cost is increases. When a firm’s product is at a higher level of environmental friendliness, then the total cost will increase. Hence, the firm tends to decrease the level of environmental friendliness, which reduces the average value in total. In addition, the socially expected environmental friendliness increases with the subjective effect of CEA. When consumers prefer more environmentally friendly products, the firms will cater for the consumers’ preferences to increase their environmental characteristics. This results in a higher average level of environmental friendliness.

(3) The two firms’ profits are the same. First, the profit is half of the price premium. Second, the profit is determined by the subjective and social effect of CEA. The profit increases with the subjective effect and decreases with the social effect of CEA because
$$\frac{\partial \pi_{i}}{\partial t}\geq 0,\;\frac{\partial \pi_{i}}{\partial d}\leq 0,\;i=1,2.$$

Therefore, the subjective effect of CEA is beneficial to the improvement of corporate profits, while the social effect of CEA inhibits the increase in corporate profits.

(4) As for the environmental characteristics of the two firms’ product, let D(t,d)=(1−b)−a be the difference between the two firms in terms of environmental friendliness. Such that
(29)$$D\left(t,d\right)=D\left(t,d\right)=\left(1-b\right)-a=\frac{3t}{2(3d+t)}.$$

It shows that the product difference increases as the subjective effect of CEA increases, which is in accordance with the results of the standard Hotelling model. However, the product difference decreases as the social effect of CEA increases.

According to the results of the standard Hotelling model, the two firms should symmetrically locate at two sides of the midpoint if we do not take the social effect of CEA into consideration. In our model, we take the average level of environmental friendliness of the products purchased by all consumers as the base level of environmental awareness of social expectations, and introduce the deviation between individual consumption and social consumption as a measure of the social effect of CEA into the utility function of consumers, thus resulting in the two firms being symmetrically distributed on both sides of the mean value, that is, they deviate from the original midpoint position. Nevertheless, the two companies have achieved a fair share of the market and profit.

In order to more clearly reflect the subjective and social effects of CEA on the product differentiation decision-making of firms, we depict their relationship in Figure 1.

In the Figure 1, given that *d* is fixed, $\bar{x}$ will move to $\hat{\theta }$ (the midpoint) as *t* increases. Meanwhile, as *D* increases with *t*, and *a* and 1−b are also symmetrically located at the two sides of $\bar{x}$, so the firms tend to move toward the right side. Therefore, the subjective effect of CEA will enable firms to enhance the level of environmental friendliness of products. When *t* is fixed, as *D* decreases with *d*, so the firms will move toward the point of $\bar{x}$. That is, the two firms tend to decrease the difference in environmental friendliness between their products. To sum up, the subjective effect of CEA can generally promote firms to enhance the level of environmental friendliness of their products. The social effect of CEA generally enlarges the differences in the level of environmental friendliness between the firms. Specifically, Firm 1 tends to reduce the level of environmental friendliness of products, while Firm 2 tends to improve the level of environmental friendliness of products.

## 5. Numerical Examples

In this section, we provide some numerical examples to display the subjective and social effects of CEA on the firms’ decision regarding price competition and production differentiation, and to also verify the ongoing theoretical results. In order to exclude the impact of the marginal cost of environmental quality, we fixed it to 1 (c=1).

### 5.1. Subjective Effect of CEA

Firstly, we demonstrate the subjective effect of CEA by varying the value of *t*, while fixing the value of *d* at some constants (e.g., 1.5, 3, 10, 20, 40, 50). Then, by analyzing the change of the products’ environmental quality difference, enterprise pricing, and profits, we show the subjective effects of CEA on the firms’ pricing and differentiation strategies.

In Figure 2, the horizontal axis is the subjective effect of CEA, and the vertical axis represents the environmental quality of the enterprise’s products and their differentiation level. In Figure 2a, the value of *d* changes from 1.5 to 50. Figure 2b,c are the minimal and maximal cases where the value of *d* is 1.5 and 50, respectively. Firstly, when the subjective effect value increases, the environmental quality level (black solid line) of Firm 1 product increases first and then decreases, and there is a maximum value, but it moves to the right as the value of *d* increases from low to high. Secondly, the environmental quality level of Firm 2’s products (red solid line) is a monotonic increasing function of the subjective effects, and it also moves to the right as the value of *d* increases. Finally, the difference of product environmental quality (blue solid line) is a monotonic increasing function of the subjective effect, which is consistent with the previous conclusion.

In the Figure 3, the horizontal axis is still the subjective effect of CEA, while the vertical axis is the price and income of the enterprise. It can be found that the product price (the red solid line for Firm 1, the blue solid line for Firm 2) and the income level (black solid line) are the incremental functions of the subjective effects of CEA, which are consistent with our previous conclusions. Furthermore, they all increase with the increase in *d*’s value. Secondly, the price of Firm 2 is always no less than Firm 1, because the environmental quality level of Firm 2’s products is higher than Firm 1’s, so according to the price composition obtained in Section 4, the price of Firm 2 should be relatively high. However, the difference between product prices decreases with the increase in *d*’s value. Therefore, this shows that the price composition of the enterprise’s products mentioned above is reasonable.

### 5.2. Social Effect of CEA

In this part, we fix the value of the subjective effects at some constants (e.g., 1.5, 3, 10, 20, 40, 50). Then, we demonstrate the social impacts of CEA on the products’ environmental quality difference, enterprise pricing, and profits.

In Figure 4, the horizontal axis is the social effect of CEA, and the vertical axis is the level and difference of environmental quality of the enterprise’s products. It can be found that the social effects of CEA have opposite effects on Firm 1 and 2. When the social effect of CEA is enhanced, Firm 1 improves its product environmental quality level (black solid line), while Firm 2 reduces its product environmental quality level (red solid line). However, the difference in the environmental quality level (blue solid line) between the two enterprises decreases with the increase in the social effect of CEA, which also verifies the relevant conclusion we derived before, that is, the social effect of CEA weakens the difference of the environmental quality of products and intensifies the competition between enterprises. In addition, it can be found that when the value of *t* increases, the enterprises improve the environmental friendliness of products, and the differences between them gradually increases as well.

In Figure 5, the horizontal axis represents the social effect of CEA, and the vertical axis represents the price of the product and the profit of the enterprise. First of all, it is noted that both product prices and corporate profits are subtractive functions of CEA’s social effects, which confirms our previous conclusions. Secondly, the product price of Firm 2 (line solid line) is always no less than that of Firm 1 (blue solid line), which proves the rationality of our conclusion on the firms’ pricing strategy. However, with the increase in *t*’s value, the difference gradually narrows down. In addition, the firms’ profits (black solid line) also increase with the value of *t*’s.

## 6. Recommendations for Price Competition and Product Differentiation

On the basis of the normal solution above, we propose some possible recommendations regarding price competition and product differentiation to firms who take the subjective and social effects of CEA into consideration.

Whether through price competition or product differentiation, the purpose is to enhance the enterprises’ profits. In this model, the firm’s profit is an increasing function of the subjective effect of CEA and a decreasing function of the social effect of CEA. Therefore, in order to increase profits, enterprises can start from two positions: improving the subjective effect of CEA or reducing the social effect of CEA.

For the former, it is noticed that when the subjective effect of CEA is enhanced, consumer surplus will decrease, which will weaken the consumers’ willingness-to-buy. Although we assume that the consumer intrinsic utility is large enough, this assumption is ultimately unrealistic. Therefore, enterprises should strengthen a reasonable concept of green consumption to consumers, that is, green consumers should pay more attention to products whose environmental quality level is equal to their environmental awareness level. When the deviation between the two is reduced, the subjective effect of CEA will not lead to a decrease in purchasing intention caused by the reduction in consumer surplus. At the same time, it is also noted that enterprise pricing is an incremental function of the subjective effect of CEA, so when the subjective effect of CEA is enhanced, the product price of enterprises will also increase, and the profits of enterprises will be guaranteed. However, the higher the firm’s pricing, the lower the consumer’s willingness to buy. Therefore, when the subjective effect of CEA is enhanced, enterprises should strengthen the research and development of green production technology to reduce the production cost of environmentally friendly units. Furthermore, when the subjective effect of CEA is enhanced, the environmental quality difference of the enterprise’s products will be enlarged. In the limited space of differentiation, enterprises can use their technological differences in green production to pursue differentiated product strategies in order to meet customers’ different environmental quality needs, thus weakening the competition between the enterprises.

For the latter, a reduction in the social effect of CEA enhances corporate profits. Firstly, when the social effect of CEA is increased, the consumers’ willingness to buy decreases accordingly. Therefore, firms should encourage consumers to approach the level of environmental friendliness expected by society to compensate for the reduction in consumer surplus caused by the reduction in the consumers’ purchase intention. Therefore, firms should strengthen the publicity and promotion of green products in order to establish the level of environmental quality recognized by society. Secondly, when the social effect of CEA is reduced, the environmental quality difference between the enterprises’ products will expand, which is beneficial to the development of the product differentiation strategy. Therefore, enterprises should strengthen the segmentation of consumers, encourage the research and development of unique green products to meet the needs of specific consumers, thereby weakening competition among enterprises.

## 7. Conclusions

CEA is an important determinant affecting the consumers’ green purchasing behavior, which provides novel opportunities for green product manufacturers, making them consider the impact of CEA on their competitive strategies. This paper breaks through the traditional way of dealing with the impact of CEA as a whole, by dividing the impact of CEA on consumers into two main parts: the subjective and social effects, and integrates them into the classic Hotelling model. On the basis of the game analysis framework, this paper analyzes the impact the subjective and social effects have on the pricing and differentiation strategies of enterprises. It is found that the subjective and social effects of CEA play an opposite function in price competition and product differentiation. The subjective effect of CEA can enhance the price and profit of enterprises, and at the same time enlarge the difference of the environmental quality of products among enterprises. Obviously, the increase in price is a guarantee of profit, because the manufacturing of green products will bring higher manufacturing costs, while the expansion of product quality level differences weakens the competitiveness of enterprises, and provides a basis for scientific customer segmentation. On the other hand, the social effect of CEA reduces the price of the enterprises’ products, reduces the profits of enterprises, and narrows the differences in environmental quality level of the enterprises’ products. Therefore, the social effect of CEA intensifies competition among enterprises. On the basis of these results, we suggest that green product producers should distinguish the effects of CEA when establishing price competition strategies and product differentiation strategies.

On the basis of the fact that the composition of CEA is complex, this paper distinguished between the subjective and social effect of CEA. The main contribution is that we theoretically clarify the opposite effect of the two different components of CEA on environmentally friendly product pricing and differentiation strategy, which demonstrates a possible reason why the effect of CEA on behavior is different, and provides a theoretical support for enterprises to price and differentiate between green products.

## Figures and Tables

**Figure 1 ijerph-17-00716-f001:**
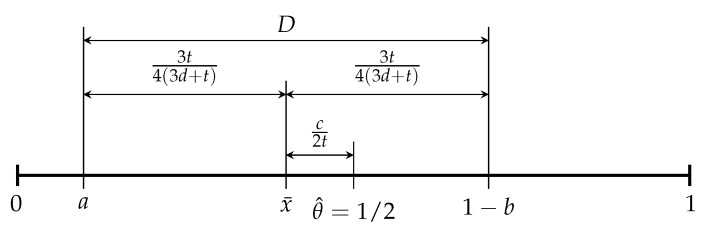
Subjective and social effects of consumer environmental awareness (CEA).

**Figure 2 ijerph-17-00716-f002:**
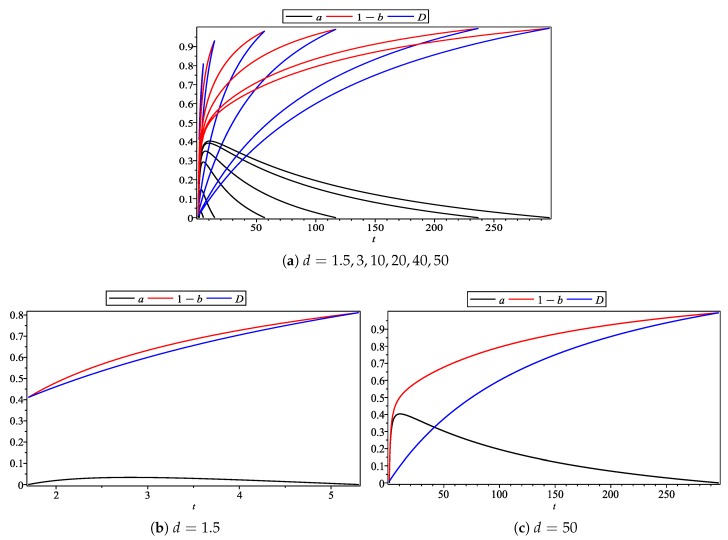
Subjective effect of CEA on the products’ environmental quality difference.

**Figure 3 ijerph-17-00716-f003:**
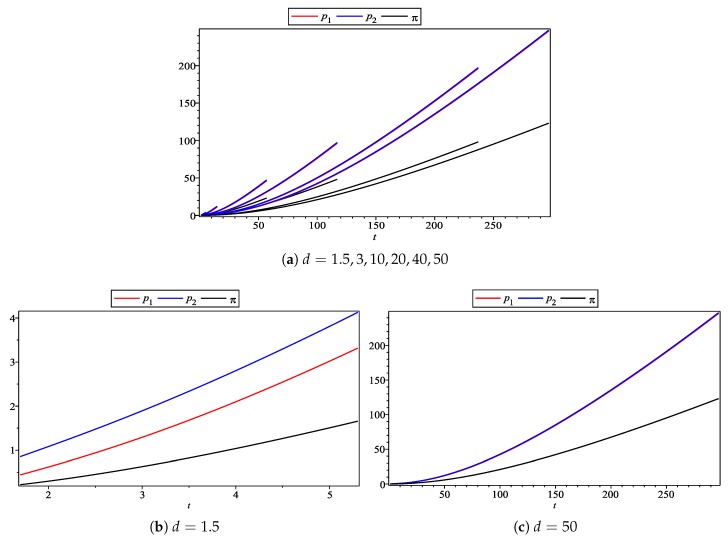
Subjective effect of CEA on enterprise pricing and profit.

**Figure 4 ijerph-17-00716-f004:**
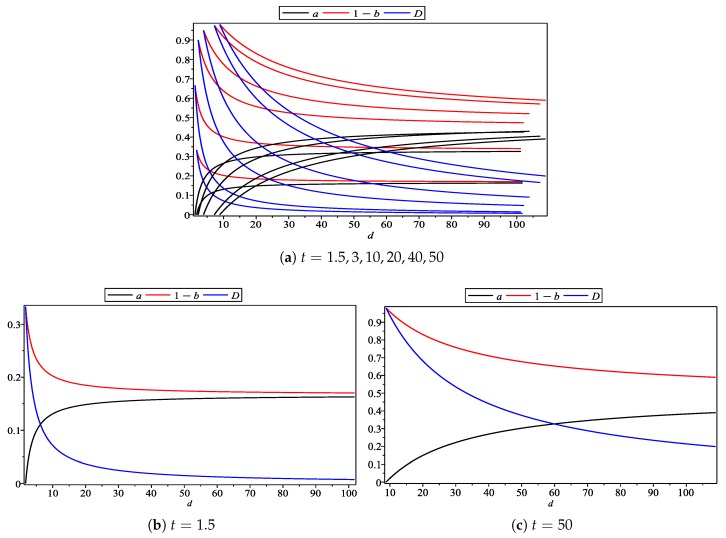
Social effect of CEA on the products’ environmental quality difference.

**Figure 5 ijerph-17-00716-f005:**
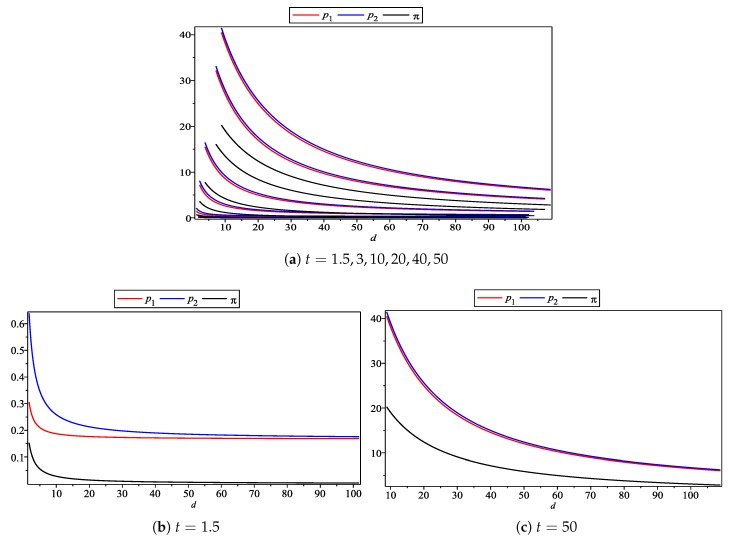
Social effect of CEA on enterprise pricing and profit.

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
