# Peer review of "Price Competition and Product Differentiation Based on the Subjective and Social Effect of Consumers’ Environmental Awareness"

_ijerph, 2020, doi:10.3390/ijerph17030716_

Round 1
Reviewer 1 Report
Please find attached my report.

Reviewer 2 Report
I suggest to reject this paper because the real cases: products, firms, are not clear. It is just theoretical assumptions, which can mean nothing. The gap between attitudes and behaviour should be discussed. The elasticity coefficient also should be included. It shows that authors are not familiar this topic.
Reviewer 3 Report
The research is well organized. The paper examines the important marketing strategies of price competition and product differentiated. i make some comments for improving.
Abstract
The abstract provide sufficient guidance, but the abstract can claim the research contribution.
Introduction
The introduction provide sufficient research background. However, I suggest that the author(s) can cite more articles including environmental awareness, subjective effect, social effect, price competition and product differentiation. Thus, this will improve research background and motivation.
The model and results.
The research design is appropriate and the methods are adequately described. Therefore, the research results is reasonable and clearly presented. However, the subjective and social effect is expected to provide more explanation for the empirical results.
Conclusions
The conclusions are supported by the empirical results.
Round 2
Reviewer 1 Report
I don't have any further comments.
Author Response
Thanks for your positive suggestions. We checked language style and grammar errors. All modifications are highlighted in blue color for your convenience.
Reviewer 2 Report
The responses of authors did not change my decision. Green purchase is very complex phenomenon. It depends on what products are examined (food, cosmetic, clothes and etc.), because people buy these products not only because they are environmentally friendly but also to other reasons (health care, brand and etc.). Furthermore, the affordability aspect is very important. I rich and poor countries the relationship between prices could differ. Authors distinguish social and subjective aspects, but in individualistic societies it can insignificantly affect the behavior and response to prices.
So it is not enough only theoretically analyse these aspects. The deep analysis there is required and I miss it in this paper.
he main contribution is that we clarify theoretically the opposite effect of two438different components of CEA on environment-friendly product pricing and differentiation strategy,439which states a possible reason why the CEA on behavior is different
he biggest possible innovation is to decompose CEA into two parts: personal and social effecthe biggest possible innovation is to decompose CEA into two parts: personal and social effect
Author Response
Thanks for your constructive comments. As you mentioned, green purchase is very complex phenomenon because it is affected by many reasons. Hence, it is hard for us to conduct empirical and theoretical study in one article.